# Correlation between BAP1 Localization, Driver Mutations, and Patient Survival in Uveal Melanoma

**DOI:** 10.3390/cancers14174105

**Published:** 2022-08-25

**Authors:** Yasemin C. Cole, Yu-Zhi Zhang, Beatrice Gallo, Adam P. Januszewski, Anca Nastase, David J. Essex, Caroline M. H. Thaung, Victoria M. L. Cohen, Mandeep S. Sagoo, Anne M. Bowcock

**Affiliations:** 1National Heart and Lung Institute, Imperial College London, London SW3 6LR, UK; 2Department of Histopathology, Royal Brompton and Harefield NHS Foundation Trust, London SW3 6NP, UK; 3Ocular Oncology Service, Moorfields Eye Hospital & St. Bartholomew’s Hospital, London EC1V 2PD, UK; 4Moorfields Eye Hospital, London EC1V 2PD, UK; 5Department of Eye Pathology, UCL Institute of Ophthalmology, London EC1V 9EL, UK; 6Departments of Oncological Sciences, Dermatology and Genetics & Genome Sciences, Icahn School of Medicine at Mount Sinai, New York, NY 10029, USA

**Keywords:** uveal melanoma, biomarkers, survival, metastasis, *BAP1*, *SF3B1*, *EIF1AX*, monosomy 3, immunohistochemistry, nonsense-mediated decay

## Abstract

**Simple Summary:**

100 uveal melanomas from the UK were analyzed for molecular biomarkers, including alterations of chromosomes 3 and 8, cellular localization of BAP1, and genes known to be mutated in uveal melanoma. Consistent with earlier studies, loss of nuclear BAP1 (nBAP1) predicted shorter overall survival. Tumors with BAP1 loss of function mutations frequently exhibited heterogeneous BAP1 staining, and tumors with ≥25% loss of nBAP1 were a more reliable prognosistic indicator than chromosome 3 loss (LOH3) or chromosome 8q gain. Regardless of mutation class, most *BAP1* mutations led to loss of nBAP1 and aberrant expression of cBAP1.

**Abstract:**

Uveal melanoma (UM) is an uncommon but highly aggressive ocular malignancy. Poor overall survival is associated with deleterious *BAP1* alterations, which frequently occur with monosomy 3 (LOH3) and a characteristic gene expression profile. Tumor DNA from a cohort of 100 UM patients from Moorfields Biobank (UK) that had undergone enucleation were sequenced for known UM driver genes (*BAP1*, *SF3B1*, *EIF1AX*, *GNAQ**,* and *GNA11*). Immunohistochemical staining of BAP1 and interphase FISH for chromosomes 3 and 8 was performed, and cellular localization of BAP1 was correlated with *BAP1* mutations. Wildtype (WT) BAP1 staining was characterized by nBAP1 expression with <10% cytoplasmic BAP1 (cBAP1). Tumors exhibited heterogeneity with respect to BAP1 staining with different percentages of nBAP1 loss: ≥25% loss of nuclear BAP1 (nBAP1) was superior to chr8q and LOH3 as a prognostic indicator. Of the successfully sequenced UMs, 38% harbored oncogenic mutations in *GNA11* and 48% harbored mutations in *GNAQ* at residues 209 or 183. Of the secondary drivers, 39% of mutations were in *BAP1*, 11% were in *EIF1AX*, and 20% were in the *SF3B1* R625 hotspot. Most tumors with *SF3B1* or *EIF1AX* mutations retained nuclear BAP1 (nBAP1). The majority of tumor samples with likely pathogenic *BAP1* mutations, regardless of mutation class, displayed ≥25% loss of nBAP1. This included all tumors with truncating mutations and 80% of tumors with missense mutations. In addition, 60% of tumors with truncating mutations and 82% of tumors with missense mutations expressed >10% cBAP1.

## 1. Introduction

Approximately 7000 individuals worldwide per year are diagnosed with uveal melanoma (UM) [1], accounting for 3.7% of all melanomas [1,2]. Although a relatively rare malignancy, UM is associated with significant morbidity, including, but not limited to, the loss of sight and ultimately the eye in advanced stages. UM arises from melanocytes of the uveal tract [3]. An estimated 85–90% of tumors arise in the choroid, while the remaining cases are confined to the iris (2–4%) or ciliary body (6–7%) [1,4,5,6]. While the 5-year relative survival rate is approximately 82%, 50% of patients develop metastases within 10–15 years of diagnosis despite successful treatment of the primary tumor [7,8,9]. After metastases, the median overall survival (OS) ranges from 4 to 15 months [10,11]. Currently approved therapies, such as monoclonal antibodies targeting CTLA-4 (Ipilimumab) and PD-1/PD-L1 axis (Pembrolizumab, Nivolumab, Atezolizumab), provide limited efficacy [11,12,13,14].

Biomarkers for metastatic risk include a characteristic gene expression profile (GEP), and the presence of monosomy chromosome 3 (LOH3) and gain in chromosome 8 [15,16,17,18]. UMs classified as class 1 on the basis of an established GEP have a low metastatic risk (2–21%), and have a good prognosis, whereas class-2 UMs, associated with LOH3, have a high metastatic risk (72%) and are associated with worse prognosis [19]. Activating mutations in proto-oncogenes encoding G protein subunit alpha q (*GNAQ*) and G protein subunit alpha 11 (*GNA11*) are early events in UM development, and do not provide prognostic value [20,21]. Inactivating mutations in BRCA1-associated protein 1 (*BAP1*) are found in approximately 40% of all UMs, and 84% of class-2 UMs [22,23,24,25,26]. *BAP1* encodes a tumor suppressor protein that is encoded on chromosome 3p21 [27]. Loss of BAP1 is implicated in a distinct subset of cancers that also include mesothelioma [28,29], clear cell renal cell carcinoma [30], and cholangiocarcinoma [31].

Immunohistochemical analysis of BAP1, and specifically, loss of its nuclear localization, has been evaluated as a prognostic marker in UM [32,33,34,35]. Here, we describe an investigation into BAP1 cellular localization in a cohort of UM patients from the United Kingdom, and correlate these findings with well-annotated clinical information and sequence alterations in the major UM driver genes. We observed, consistent with other studies, that irrespective of mutation class, most mutations led to loss of nBAP1 and an increase in cytoplasmic localization of BAP1 (cBAP1). We also provide evidence from TCGA that some tumors carrying truncating mutations in *BAP1* escape nonsense-mediated decay (NMD), which could account for increased levels of cBAP1. We show that ≥25% loss of nBAP1 is a reasonable cutoff for predicting shorter OS.

## 2. Materials and Methods

### 2.1. Patient Tissues

Tumor samples from 100 primary enucleations were acquired retrospectively from Moorfields Biobank (London, UK) and other collaborating centers. Moorfields Hospital treats 250–300 melanoma cases a year, and about 1/3 of those have enucleation. As a retrospective study, informed consent was obtained from patients prior to surgery for future research. The study was conducted according to the principles of the declaration of Helsinki and Human Tissue Act. Ethical approval was granted from the Institutional Review Board at Imperial College London and Moorfields Eye Hospital (reference 10/H0106/57-2014ETR37). All included patients received enucleation between March 2012 and December 2013 at the London Ocular Oncology Service (St. Bartholomew’s Hospital and Moorfields Eye Hospital) with a subsequent histological diagnosis of UM. Formalin-fixed paraffin-embedded (FFPE) tumor sections were obtained from Moorfield’s Biobank. Clinical data (e.g., date of primary tumor diagnosis, metastasis, date of last follow-up) and pathological information (e.g., cytogenetic results of interphase fluorescence in situ hybridization (FISH)) from each patient were retrieved from the biobank and collaborating institutions.

### 2.2. DNA Isolation and Quantification

To prepare the tissue samples for molecular analysis, genomic DNA was isolated from the FFPE tumor sections. Genomic DNA was extracted and purified with the QIAamp DNA FFPE Tissue Kit (ca. no. 56404, Qiagen, Hilden, Germany) according to the manufacturer’s instructions, and quantified with the Quant-iT^TM^ PicoGreen dsDNA Assay Kit (ca. no. P7589, Thermo Fisher Scientific, Waltham, MA, USA). Due to the lack of DNA, we were unable to assess chromosome 3 and 8q status via MLPA (Multiplex Ligation-dependent Probe Amplification) or comparable methods, and thus, relied on clinical interphase FISH data. 

### 2.3. Primer Design and Polymerase Chain Reaction Amplification

Primers were designed to amplify *BAP1* gene coding exons, exon 1 and 2 of *EIF1AX* gene, and a region of exon 14 of *SF3B1* gene that encodes the recurrently mutated R625 residue. Primers and conditions for PCR amplification are shown in Appendix A. There was marked variation in the DNA concentration of the samples, ranging from 2.2 ng/µL to 339 ng/µL, and five samples could not be sequenced due to insufficient DNA quantity (UM #0858, #1712, #2302, #2847, and #3019). However, Sanger sequencing was successful in all 95 remaining tumors. Three samples (UM #867, #2611, #1303) were only sequenced for *BAP1* exons 3–13, exons 3–4 and 6–14, and exons 3–4 and 6–8, respectively, due to low DNA quantity. 

### 2.4. Sanger Sequencing and Bioinformatic Analysis of Genetic Variants

Amplified DNA was subjected to Sanger Sequencing by Eurofins Genomics, Germany, and analyzed with Sequencher (version 5.3, Gene Codes). Since matched normal samples were not available, unknown variants were queried against the exome aggregation consortium (http://biorxiv.org/content/early/2015/10/30/030338) (accessed on 1 August 2017) release 0.3 and Kaviar (version 160204-Public) [36]. Novel variants with an MAF (minor allele frequency) >10^−5^, and common single-nucleotide variants, were filtered out to exclude non-predisposing mutations. Pathogenicity of novel variants was determined with Variant Effect Predictor (Ensembl CRCh37 release 89) [37]. 

Visualization of the consequences of *BAP1* mutations on *BAP1* mRNAs was achieved with TCGA datasets and the Integrative Genomics Viewer (IGV) (https://software.broadinstitute.org/software/igv/) (accessed on 3 June 2021) [38]. The oncoprint figure was generated based on the package developed by Daniel Klevebring on his github: https://github.com/dakl/oncoprint (accessed on 27 July 2022). The lollipop plot was generated following the instructions provided by the developers of trackViewer (Dr. Jianhong Ou and Dr. Lihua Julie Zhu) within Bioconductor. Both the oncoprint and lollipop images were redrawn with the locations of known domains mapped onto the gene. 

### 2.5. Immunohistochemistry

Staining of BAP1 was performed on deparaffinized 4-µm FFPE tissue sections with the Ventana Benchmark Ultra platform (Ventana Medical Systems Inc., Oro Valley, AZ, USA). Antigen retrieval was performed with Ventana CC1 buffer with an amplification kit at 100 °C for 64 min, and anti-BAP1 C-4 monoclonal antibody raised to amino acids 430–729 of human BAP1 (cat no. sc-28383; Santa Cruz Biotechnology Inc., Dallas, TX, USA) applied at 1:50 dilution at 37 °C for 12 min. This has been validated with siRNA knockdown of BAP1 in cell lines [39], and has also been used successfully in studies similar to those described here [40,41,42].

Staining was visualized with the Ventana Ultraview Alkaline Phosphatase (AP) Red Kit (Ventana Medical Systems Inc., Oro Valley, AZ, USA). Sections of human small bowel epithelium served as a positive control, and were used for each run. Immune cells were used as the internal positive control. 

Microscopic assessment was performed with a Nikon Eclipse Ci-L microscope (Nikon Corporation, 1 The Crescent, Surbiton KT6 4BN, Surrey, UK) and a field area measuring 0.24 mm^2^ per high-power field (HPF, ×400 magnification). Microscopic images in 300-dpi TIFF format were taken from representative cases using a Nikon Digital Sight DS-L3 camera (Nikon Corporation).

Intact “positive” BAP1 expression was defined as nuclear staining within cancerous cells, and loss “negative” BAP1 expression was defined as >50% loss of nuclear immunostaining within cancerous cells, utilizing immune cells as an internal positive control. 

Blinded to sequencing results and patient outcome, pathologists (Y.Z.Z. and C.M.H.T., with no prior knowledge of chromosomal nor genetic status) scored BAP1 staining by 5% increments for the presence of protein expression and its localization across the entire tumor. The interobserver agreement between the two scorers (Y.Z.Z., C.M.H.T.) regarding BAP1 immunohistochemistry was assessed with Cohen’s κ. The strength of agreement was categorized as follows: poor (<0.20), fair (0.21–0.40), moderate (0.41–0.60), good (0.61–0.80), excellent (0.81–1.00). All statistical analyses were performed using SPSS 26 (IBM Corp., Armonk, NY, USA).

Cutoff levels were determined by the minimum percentage loss of tumor cell BAP1 nuclear staining (nBAP1). Hence, our scoring system was a two-tier system where a single cutoff threshold was used. This differed from that used by Stalhammer et al. [43]. A predominant pattern corresponded to a cutoff threshold of 50% loss of nBAP1 (equivalent to score 0–1, and in some cases score 2, described elsewhere [43]). A 25% cutoff was equivalent to score 0–2, and in some cases score 3, described elsewhere [43]). A direct comparison of thresholds is not possible, but our predominant pattern is the closest to that described by Stalhammer et al. [43].

### 2.6. Statistical Analysis

To evaluate the effect of altered cellular localization, chromosomal changes, and mutation status of *BAP1*, *SF3B1*, and *EIF1AX* on patient prognosis, Kaplan–Meier (KM) analyses of overall survival (OS) were performed [44]. The Cox–Mantel log-rank test was performed to determine whether the groups plotted on the KM analysis were significantly different, where *p* < 0.05 was deemed statistically significant [45]. Cases with missing data were excluded from statistical analysis with respect to the specific variable. The *p* values of the statistical analyses were adjusted with the Bonferroni correction when there were multiple comparisons. Patients were censored if death did not occur by the time of last follow-up, or was a result of another cause. Categorical variables were analyzed with Fisher’s exact and chi-square tests.

## 3. Results

### 3.1. Demographics and Clinical Characteristics

We investigated 100 uveal melanomas treated by primary enucleation at a tertiary ocular oncology center in the UK (Moorfields Eye Hospital and St. Bartholomew’s Hospital). The patient demographic and clinical characteristics were comparable to published studies (Appendix A). The median age at diagnosis was 67.5 years (range 23–96 years; mean age 67 years). Stage IIIA–IIIC disease (AJCC TNM staging system, 8th edition [46,47]) represented 47% (47/100) of cases. Similar to previous studies of UM, 67% of tumors involved only the choroid, and 27% also involved the ciliary body and the iris. Survival data were available for 96 patients (96%), of which 52 (54.2%) had died at the time of analysis (August 2019). The median OS of patients was 58.8 months (95% CI 44.7–2.9), with an estimated 5-year survival of 48.1%. Thirty-four patients (35.4%) subsequently developed metastatic disease, with a median time of 20.2 months from the initial diagnosis (95% CI 0–74.5). 

### 3.2. Mutations in Driver Genes

We attempted DNA isolation and Sanger sequencing of 95 FFPE-derived tumor specimens. Sequencing targets were: mutation hotspots in *GNAQ* and *GNA11* (residues 183 and 209); coding exons 3–17 of *BAP1;* exon 1 and 2 of *EIF1AX* [24,48], and exon 14 of *SF3B1*, which harbors the codon encoding its recurrent R625 alteration [49]. A list of all *BAP1*, *EIF1AX*, and *SF3B1* mutations identified, along with corresponding clinical details are described in Appendix A. An oncoprint summarizing mutations in these genes in our cohort is shown in Figure 1A.

Of the successfully sequenced Ums, 38% harbored oncogenic mutations in *GNA11*, and 48% harbored mutations in *GNAQ*. The presence of *GNA11* and *GNAQ* activating oncogenic mutations at residues 209 or 183 were mutually exclusive, as expected. Of the secondary drivers, 39% of mutations were in *BAP1*, 11% were in *EIF1AX*, and 20% were in the *SF3B1* R625 hotspot. The majority of the *BAP1* mutations were novel, and are shown in the lollipop plot on Figure 1B, while *SF3B1* mutations have been reported elsewhere [49]. Two tumors harbored mutations in two driver genes (UM #2651 (*BAP1* and *SF3B1*) and #1064 (*BAP1* and *EIF1AX*)). *EIF1AX* mutations affected the N terminus, as previously reported [24], and included p.Pro2Ser/Leu/Arg, p.Asn4Ser/Thr, and p.Lys5Glu. Overall, 31% of tumors had no detected secondary driver mutations, which was lower than expected, and was likely due to technical challenges in sequencing DNA isolated from FFPE samples. 

### 3.3. Correlation of BAP1 Mutations with Its Altered Subcellular Localization

Ninety tumors were successfully subjected to immunohistochemical analyss: Tumor sections were labeled with an antibody to BAP1, and protein abundance and cellular localization were determined. The overall interobserver agreement between two scorers regarding BAP1 immunohistochemistry, based on the predominant staining pattern, was moderate to good (Cohen’s κ 0.709, 95% CI 0.574–0.844, *p* = 5.1 × 10^−13^).

Wildtype (WT) BAP1 staining is characterized by nBAP1 expression, with or without cytoplasmic BAP1 protein (cBAP1) expression (<10%, see below). Tumors exhibited heterogeneity of BAP1 staining with different percentages of loss of nBAP1, as reported by others [41,50]. There were three distinct staining patterns in UMs: wildtype, cytoplasmic-only, and total loss of staining. Figure 2A shows these representative BAP1 staining patterns. A summary of percentages of nBAP1 versus loss of nBAP1 in tumor cells grouped according to mutation class is depicted in Figure 2B. 

Loss of nBAP1 expression as the predominant staining pattern (>50% of tumor cells) was observed in 55/90 (61%) tumors, and the majority of these were LOH3 (Figure 2B, Appendix A). The majority (24/26) of tumor samples with likely pathogenic *BAP1* mutations displayed loss of nBAP1 (Figure 2B, Appendix A). These included all 15 tumors with truncating mutations. Nine of these tumors (60%) also expressed cBAP1 at values greater than 10% of cells, and greater than that seen with WT *BAP1*. Nine of the eleven tumors with missense mutations also displayed loss of nBAP1 (81.8%), and all of these expressed cBAP1 at values greater than seen with WT *BAP1*. Tumors with *SF3B1* or *EIF1AX* mutations exhibited 0–10% cBAP1 (Appendix A). Hence, >10% cBAP1 is likely to be indicative of pathogenicity. A tumor with a splicing mutation in *BAP1* had also undergone loss of nBAP1, but had WT levels of cBAP1. 

We were unable to detect germline mutations in *BAP1* due to our study design, whereby we did not have matched normal DNA of patients. 

There were two tumors with LOH3 as well as retained WT BAP1 staining, but no identified driver mutation (UM #1658 and UM #2847). UM #1658 had developed metastases. *MBD4* maps to chromosome 3 and is a predisposing tumor suppressor gene for UM associated with hypermutated tumors with LOH3 [51]. However, due to lack of material, it was not possible to test these samples for *MBD4* mutations. 

### 3.4. Prognostic Impact of Biomarkers

LOH3 is historically the chromosomal event most strongly associated with UM metastasis [52,53,54,55]. However, a gain in chromosome 8q was detected in over 70% of UMs [56], and its co-occurrence with LOH3 is associated with worse prognosis than LOH3 alone [47,52,57,58,59,60,61]. Of the 100 patients in the current study, 70 patients had cytogenetic testing as part of their clinical management. The vast majority (98.6%, n = 69) of patients had cytogenetic testing of both chromosome 3 and chromosome 8q. In total, 53.6% (37/69) of tumors were LOH3, and 69% (48/69) had chromosome 8q gain. Furthermore, 47.8% (33/69) of the tumors were both LOH3 and had a gain in chromosome 8q; 54.5% (18/33) of these patients had died at the time of analysis (Appendix A). 

In keeping with the literature, the pathological stage was predictive of OS (Appendix A). Age (Appendix A) and sex (not shown) were not significant prognostic variables. Consistent with earlier findings, LOH3 conferred a worse prognosis (Appendix A) (*p* = 2.5 × 10^−5^), and there was a significant trend with chromosomal 8q gain (*p* = 0.013) (Appendix A), as reported previously. Loss of nBAP1 as the predominant staining pattern was prognostically significant (*p* = 1.5 × 10^−4^). The presence of *BAP1* mutation also conferred a worse prognosis (*p* = 0.003), but did not perform as well as chromosome 3 status or BAP1 IHC (Appendix A). This may be due to poor quality DNA from some of the FFPE specimens. BAP1 IHC exhibited satisfactory sensitivity (87.2%) and specificity (80.0%) in predicting underlying *BAP1* mutation or chromosome 3 aneuploidy (Appendix A).

### 3.5. Prognostic Impact of Percent Loss of nBAP1 Expression

We further explored the optimal threshold for levels of loss of nBAP1 expression in relation to its prognostic impact by applying cutoffs of nBAP1 at 1–49% (Figure 3A), 10–49% (Figure 3B), and 25–49% (Figure 3C). We then estimated overall survival on the basis of each of these cutoffs. These yielded *p* values of 0.55, 0.49, and 0.004, respectively. A value of ≥25% was determined to be a reliable cutoff based on the prognostic impact of 25–49% loss of nBAP1, and was similar to a value of ≥50% loss of nBAP1 (*p* = 9 × 10^−6^). 

### 3.6. Nonsense-Mediated Decay May Lead to an Increase in cBAP1 in Tumors with Truncating Mutations of BAP1

*BAP1* truncating mutations are expected to lead to loss of both nBAP1 and cBAP1 via nonsense-mediated decay (NMD). NMD is the translation-dependent degradation of mRNAs harboring premature termination codons (PTCs). In mammalian cells, NMD is generally splicing-dependent, and usually requires that a PTC be localized at least 50–55 nt upstream of an exon–exon junction [62]. However, some truncating mutations can escape NMD, and we observed a substantial fraction of tumors with *BAP1* truncating mutations that exhibited more cBAP1 than seen in WT *BAP1* tumors (Appendix A). This suggests the evasion of NMD or other processes, such as exon skipping. We investigated which of these options was likely to have occurred in uveal melanoma and other cancers with *BAP1* mutations. To do this, we interrogated TCGA datasets where data of both mutation and RNA sequencing were available. We first examined *BAP1* mRNA levels in tumors with mutant and WT *BAP1*. In some UMs with truncating mutations, the level of BAP1 mRNA was low, indicative of NMD (e.g., Appendix A, tumors A985 and A9EV). However, other tumors exhibited a range of levels of *BAP1* mRNA, which were sometimes as high as those seen in *BAP1* WT tumors (Appendix A, e.g., tumors A88A and A9F1). We then visualized *BAP1* transcripts from TCGA datasets with the Integrative Genomics Viewer (IGV), and confirmed the presence of *BAP1* transcripts in tumors with truncating mutations and high levels of *BAP1* mRNA, indicating escape from NMD (Appendix A, tumors A9F1 and AB3K (mRNA levels were not available for this tumor, but it harbors an out-of-frame deletion leading to a frameshift)). Even cases with low levels of *BAP1* mRNA had often retained mutant *BAP1* transcripts, indicating partial NMD escape. In a few UMs, transcripts appeared to have been generated with a cryptic splice site within an exon, or to have undergone skipping of the exon with the truncating mutation (Appendix A, tumor A9EE). 

## 4. Discussion

Here we describe the molecular characterization of a cohort of primary UMs from 100 subjects from the UK. We performed mutational analysis of previously identified driver genes *GNAQ, GNA11, BAP1, SF3B1*, and *EIF1AX*, and chromosome 3 and 8 status, with interphase fluorescence in situ hybridization. The prognostic impact of these biomarkers was then evaluated. In keeping with the literature, pathological stage was predictive of OS. Consistent with earlier studies, we also observed a significant trend in OS with chromosomal 8q status, LOH3, and the presence of *BAP1* gene mutation. Loss of nBAP1 also conferred worse prognosis, consistent with prior studies [32,33,34,35,63]. We determined that a cutoff of ≥25% loss of nBAP1 was similar to predominant loss of nBAP1, and is therefore likely to have a suitable prognostic impact. Moreover, therapeutic choices such as the use of HDAC inhibitors, which have been proposed for BAP1 mutant tumors, might also be considered on the basis of this level of loss of nBAP1 [64].

In the current study, we show that all forms of *BAP1* deleterious mutations, i.e., truncating, missense mutations, and splicing, led to the loss of nBAP1 expression, consistent with other studies [34,35,65]. However, as in an earlier study, tumors exhibited heterogeneous BAP1 expression [34]. Tumorigenesis is not expected to occur without the loss of BAP1 function in the majority of class-2 tumors, thus, cells expressing nBAP1 are more likely to be non-tumor cells, such as macrophages or melanophages.

Of the 55 UMs with loss of nBAP1, metastasis status was known for 54 patients; of these, 30 had undergone metastasis (55%). This is similar to a study of 30 enucleated eyes, where 68% of tumors with low nBAP1 and 55% of tumors with high cBAP1 developed metastasis [40]. Farquhar and colleagues [65] reported that most frameshift *BAP1* mutations led to cytoplasmic BAP1 staining, and in the current study, nine of fifteen UMs with truncating mutations were associated with the presence of cBAP1.

The role of WT cBAP1 in the cell is of interest. Although it is found at very low levels, several functions have been proposed. BAP1 is a component of the polycomb repressive deubiquitinase (PR-DUB) complex [66]. PR-DUB complexes catalyze the removal of monoubiquitination on lysine 119 of histone H2A (H2AK119ub1) through a multiprotein core comprised of BAP1, HCFC1, FOXK1/2, and OGT in combination with either ASXL1, 2, or 3 [67]. PR-DUB, by counteracting the accumulation of H2AK119ub1, maintains chromatin in an optimal configuration, ensuring the expression of genes important for general functions such as cell metabolism and homeostasis [67]. In this way, loss of BAP1 can have a global effect on gene expression. Germline mutations in *BAP1* can lead to a variety of cancers [68], and loss of BAP1 has also been proposed to induce aerobic glycolysis (the Warburg effect) in such carriers [69]. Cells with reduced levels of BAP1 are also reported to exhibit decreased DNA repair by homologous recombination [70]. BAP1 is regulated by ubiquitination through the E2 ubiquitin-conjugating enzyme UBE2O, which results in its sequestration in the cytoplasm [71]. This is counteracted through TNPO1 binding [72]. Moreover, cBAP1 is reported to regulate processes via its cytoplasmic fraction, such as apoptosis via modulation of IP3R3-mediated ER Ca^2+^ release [25]. In addition, cBAP1 has been shown to interact with all subunits of the heptameric coat protein complex I (COPI) that are involved in vesicle formation, and protein cargo binding and sorting [73]. Hence, WT nBAP1 and cBAP1 may have different functions, however, in the context of loss of nBAP1 due to protein truncation, it is likely that retained cBAP1 is nonfunctional. 

The cBAP1 in most nBAP1-negative UMs in an earlier study [65] was reported to be “predominantly diffuse” with a distinct “focal perinuclear” expression pattern localized immediately adjacent to the cis Golgi network. In the current study, the staining patterns in the UMs with elevated cBAP1 were either dot-like or mixed, in contrast to the diffuse pattern seen with most WT BAP1 tumors. Whether the cBAP1 seen in the current study corresponds to the focal perinuclear expression pattern described earlier [65] requires further investigation. However, in the earlier study, the “focal perinuclear” expression correlated with truncating mutations in *BAP1*, whereas in the current study, the dot-like pattern was seen in tumors with truncating mutations, but also in some tumors with missense mutations in *BAP1*. This might be consistent with the accumulation of misfolded proteins in the endoplasmic reticulum, and failure to execute apoptosis [74]. 

We also provide evidence that the elevated cBAP1 found in some tumors with truncating mutations is likely due to the escape from nonsense-mediated decay (NMD) of *BAP1* mRNA. There is already a precedent for this occurring in some tumor suppressor gene mRNAs. For example, we evaluated a study of exome and transcriptome data from 9769 human tumors reported elsewhere [74]. This revealed that 30% of deleterious BAP1 mutations escape NMD [75] (ibid., Appendix A), which supports our hypothesis. Most of these tumors would have lost the nuclear localization signal of BAP1, which resides at the C-terminus and has been mapped to residues 699–729 [71], accounting for the retention of BAP1 in the cytoplasm. 

## 5. Conclusions

A cohort of 100 uveal melanomas from the United Kingdom was profiled for biomarkers, including BAP1 immunostaining, *BAP1* sequence variants, LOH3, and chromosome 8q gain. A ≥ 25% loss of BAP1 nuclear staining was a reliable indicator of BAP1 loss. The existence of cBAP1 in the presence of truncating mutations and nBAP1 loss may be due to escape from NMD. 

## Figures and Tables

**Figure 1 cancers-14-04105-f001:**
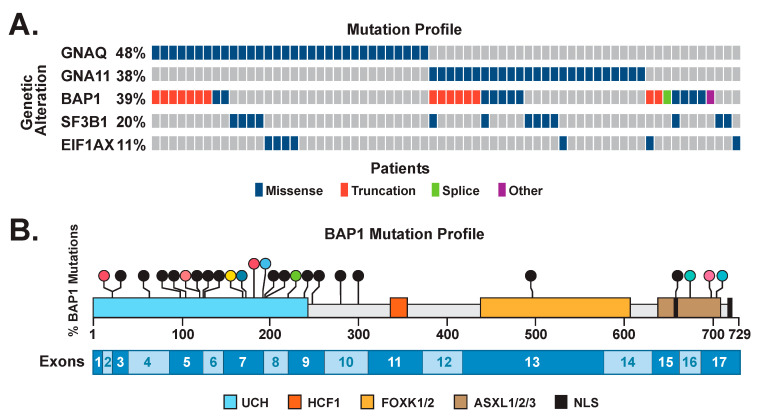
(**A**) Oncoprint of driver mutations in uveal melanoma. Individual genes are represented as rows, and individual cases or patients are represented as columns. The mutation type is shown. (**B**) Lollipop plot depicting the locations of *BAP1* mutations in the cohort. Missense mutations are depicted in color. Truncating mutations are depicted in black. The single splicing mutation is depicted in green. The UCH domain and nuclear localization signals are shown, along with regions interacting with HCF1, FOXK1/2, and ASXL1/2/3. The locations of coding exons are shown at the bottom of the figure.

**Figure 2 cancers-14-04105-f002:**
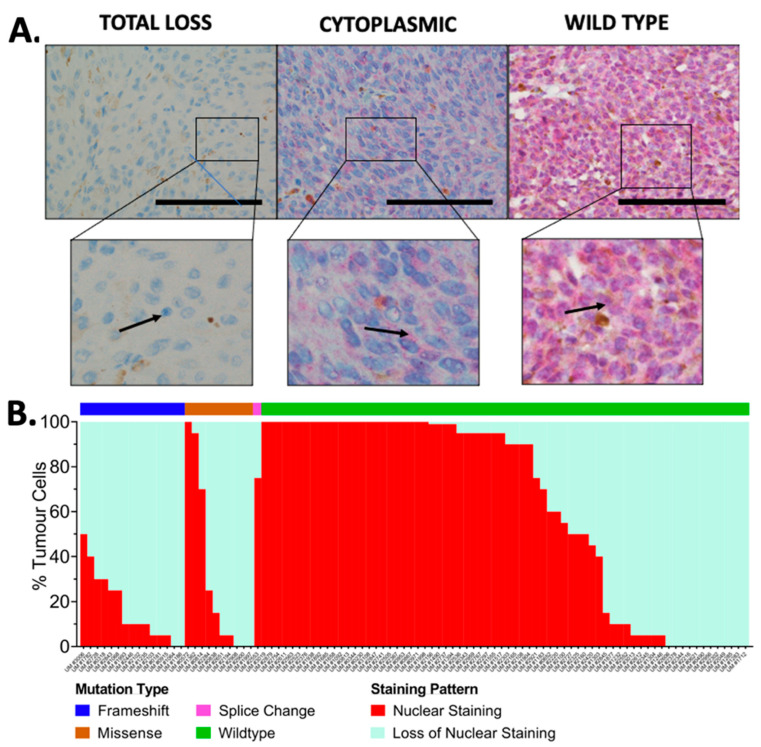
(**A**) BAP1 Immunohistochemical staining illustrating total loss of nuclear staining, cytoplasmic staining, and WT staining patterns. Immunohistochemical staining of upper images is at a magnification of 400×. The scale bar corresponds to 100 µm. Arrows point to location of BAP1 in the expanded inserts. Tumors were the following: total loss (UM #1064: *BAP1*:p.Val99CysfsTer3); cytoplasmic (UM #1235: *BAP1*:p.Leu248fs*1); wildtype (UM #1892: *BAP1* WT and *SF3B1*:p.Arg625Leu). (**B**) BAPI IHC staining according to BAP1 mutation type. The percentage of tumor cells expressing nBAP1 or loss of nBAP1 were plotted based on *BAP1* mutation type (frameshift/truncating, missense, splice change, or wildtype). Note: UM #2553 harbored a *BAP1* splice and missense variant, and is represented once as containing a splice change.

**Figure 3 cancers-14-04105-f003:**
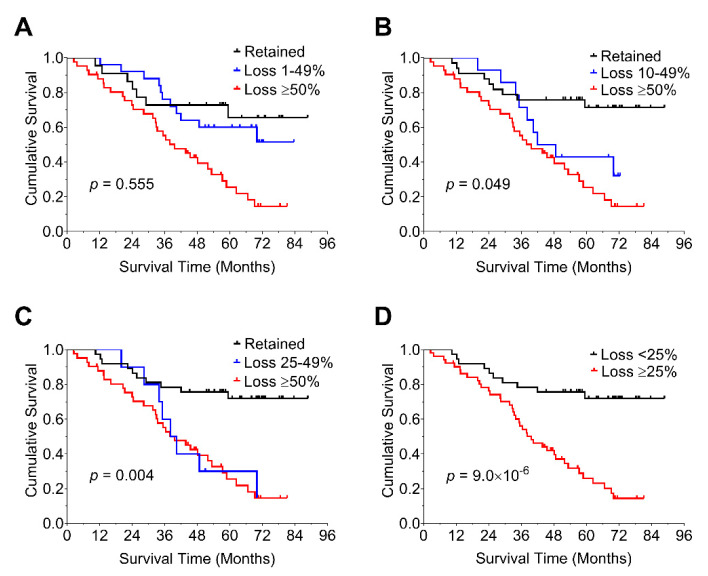
Percent loss of nuclear BAP1 staining in relation to its prognostic impact. Kaplan–Meier estimates of overall survival are graphed based on the extent of loss of nBAP1 staining using the cut offs of 1–49% (**A**), 10–49% (**B**), and 25–49% (**C**) (n = 90). A value of ≥25% was determined to be a reliable cutoff based on the prognostic impact of 25–49% loss of nBAP1 (**D**). Cox–Mantel log-rank tests were performed to determine statistical significance (*p* < 0.05).

## Data Availability

Data collected and/or analyzed for this study are available upon request.

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
