# Peer review of "Correlation between BAP1 Localization, Driver Mutations, and Patient Survival in Uveal Melanoma"

_cancers, 2022, doi:10.3390/cancers14174105_

Round 1
Reviewer 1 Report
This is a significantly improved manuscript. The authors have been responsive to the previous review.
Author Response
Thank you for the positive feedback on our manuscript revisions
Reviewer 2 Report
The paper could be published after the modifications done by the authors. The major concern regarding the use of commercial antibodies is clarified in the material and method.
Author Response
Thank you for the positive feedback on our manuscript revisions
This manuscript is a resubmission of an earlier submission. The following is a list of the peer review reports and author responses from that submission.
Round 1
Reviewer 1 Report
This project shows a correlation between overall patient survival and BAP1 mutation, loss, and localization in a set of 100 ulveal melanomas. It concludes that loss of nuclear BAP1 correlates with poorer survival.
There are some technical issues to be clarified. The study depends on one antibody against BAP1 from Santa Cruz Biotechnology. How was this antibody validated? How was the specificity evaluated? Are there previous published studies using this same antibody that did the validation for us? There are multiple ways to check an antibody from Western blots to loss of signal after knockdown. Something needs to be reported by data or reference that addresses this. Many commercial antibodies prove to be inadequate for specific protein quantification. Do other antibodies purporting to detect BAP1 produce similar results?
The tissue levels and localization of BAP1 in this study are quantified by subjective microscopy observation. There seem to be two observers. How good is the inter-observer correlation for the tumor set?
There is not much context or background presented for this work. Actually, in this case more context would make this a more interesting set of results. Not much of the very large literature on BAP1 function in cancer is cited, even about the use of BAP1 as a prognostic indicator. What is BAP1 function in the nucleus? How could loss of that function drive tumor progression? Is there an established BAP1 function in the cytoplasm? Does BAP1 have nuclear localization sequences (NLSs) and where are they? Since BAP1 has two known NLSs near the C-terminus this could explain to the reader the results on truncated forms. Marking the NLS sites on the cartoon gene of Figure 1 would be helpful. Germline mutations in BAP1 predisposing patients to some tumors have been reported. Are they present in these patients? BAP1 is reported to be a regulator of expression of proteins involved in chromatin remodeling (and cancers); are there correlations in UM datasets? Do the results here suggest therapeutic choices? It has been proposed, for example, that HDAC inhibitors might be therapeutic but only for unspecified subsets of patients with loss of BAP1 nuclear function.
Nonsense-mediated decay is remarked on as a mechanism for BAP1 reduction for truncating mutations. This should be explained to the reader, stop codons sufficiently upstream of an exon junction and so forth.... This explanation supports the authors hypothesis and that is confidence building. The Graphical Abstract does not show this mechanism but suggests normal protein expression of shorter protein forms from shorter mRNAs?
Methods including primer sequences and PCR conditions should be listed in sufficient detail to reproduce the work and not “available upon request” as in Line 123.
The micrographs of Figure 2 were not very helpful because it is difficult to see detail like nuclear vs cytoplasmic localization. Perhaps, higher magnification views could be added or larger versions added to the supplement. And BTW, some of the text in this figure is too small to read. Similarly, critical text in Figure 3 is too small.
Reviewer 2 Report
Dear Authors,
your manuscript Correlation between BAP1 localization, driver mutations and patient survival in uveal melanoma is well written and the title is appealing.
In your manuscript, you describe a cohort of 100 patients with melanoma whom you have studied firstly for their genetic alterations in the genes GNAQ, GNA11, SF3B1, EIF1AX, and BAP1, as well as LOH3 and Gain of 8q. In addition, you examined the status of BAP1 in the nucleus and cytoplasm. With the aim to investigate the prognostic relevance of these genetic markers in terms of survival. And you could clearly show that tumor stage, status of Chr3, Chr8q and nuclear loss of BAP1 or pathogenic mutations of BAP1 correlate with worse prognosis.
Unfortunately, this is already known and published. Meaning except for the fact that focal loss of nBAP1 is significant at a cutoff of 25%, the manuscript lacks scientific novelty. Furthermore, the genetic data in particular are unusually presented in Figures 1 and 2 and are therefore difficult to understand. It would have been better and more comprehensible to summarize the genetic data in an Oncoplot (PMID: 30341162) or OncoPrinter (https://www.cbioportal.org/visualize), so that one can see at a glance which genetic alterations occur together in a patient. Incidentally, the same is true for plotting genetic alterations in the gene BAP1 (see lollipop plot and MutationMapper, references above). Furthermore, the results on the prognostic capacity of the genetic biomarker BAP1 confirm the previously published data of Farquhar, Neil et al. "Patterns of BAP1 protein expression provide insights into prognostic significance and the biology of uveal melanoma.". Also you did not provide any new insghts into the the presence of cytoplasmic BAP1 in the absence of the protein in the nucleus and the role of the localization of the protein at all. I was also confused to find a short excurse into results of an analysis of the entire TCGA cohort regarding the localization of BAP1 in the discussion, starting line 388 page 10. In summary, your manuscript lacks scientific novelty, unfortunately.
Reviewer 3 Report
Cole, et al. investigated BAP1 subcellular localization and mutational status in 100 cases of uveal melanoma. They found that BAP1 mutation resulted in loss of nuclear localization of the BAP protein and poor prognosis. Analysis of a large number of cases with this rare disease is valuable and the conclusion is clear. Basically, I think that this study will serve for clinical practice as well as further investigation of uveal melanoma. However, I have some concerns about the process from data presentation to drawing conclusion.
- (Graphical Abstract) It is depicted that BAP1 mRNA, which they describe mutant truncated, is shorter than WT mRNA. Although mutant mRNA may have frameshift or nonsense mutation, its length must be almost as long as WT mRNA. It is translated mutant BAP1 protein that may be shorter than WT, if mutant mRNA escapes nonsense mediated decay.
- (ll.92-93) All included patients received enucleation between March 2012 and December 2013 at ... : I can hardly believe that 100 cases of this rare disease was collected in a single institute only within one year and 10 months. Please confirm that this description was right.
- (l.107) What is MLPA? Please spell out.
- (l.131) What is MAF? Does this mean minor allele frequency? Please spell out.
- (ll.214-215) 1064 is represented twice ... contains frameshift and splice variant: 1064 contains two frameshifts. It is 2553 that contains frameshift and splice variant.
- (ll.221-222) Loss of BAP1 ... 55 (57%) tumors (Figure 1B): Please specify how many tumors were subject to immunohistochemical analysis. Figure 1B does not demonstrate this content. In addition, it is described that loss of BAP1 was noted in 57/90 (33 + 57) cases in Figure 3E, although it is described that (n=96) in the Figure legend. Please correct either of them or both.
- Diagnostic criteria of BAP1 immunohistochemistry is difficult to understand. Does loss of nBAP1 expression (l.221) mean > 50% loss of nuclear staining? Please specify.
- (Figure 2, Upper) These figures are too small to appreciate the immunostaining results. Higher magnification will be desirable. In addition, is the scale bar displaying 100 um right? If it is right, nuclear diameter of melanoma cell is less than 10 um and smaller than neutrophil whose nucleus is around 15 um. I think that malignant cells usually have more than twice as large nucleus as neutrophil.
- (Figure 2, Lower) Is Y-axis, % Tumor Cells, different from Y-axis, % Nuclear staining, in Figure 1B? Please explain the difference clearly. One case with missense mutation reveals 100% nuclear staining in Figure 1B, while there is no case with missense mutation that shows 100% tumor cells.
- (l.244) The majority (24/28) of tumor samples with BAP1 mutations: According to Figure 1B, 1064 and 2553 have double mutations. Isn’t the number of BAP1-mutated tumor 26?
- (Figure S2) Terms used in this figure are confusing and difficult to understand. What does “Retained” and “Loss” in Figure S2A, B, and C means? What is the difference between “Focal loss” and “Loss” in Figure S2A, B, and C? Please specify. In addition, what is the difference between “Focal loss > 25%” in Figure S2C and “Loss > 25%” in Figure S2D? Please specify. Despite the description (C) (n=96) in the Figure legend, the number of analyzed cases must be 90 (38 + 10 + 42) based on the Figure 1C itself.
- (Table S4) > 10% cBAP1: A rationale for this cut-off is not explained.